# Identification of Anti-gp41 Monoclonal Antibodies That Effectively Target Cytotoxic Immunoconjugates to Cells Infected with Human Immunodeficiency Virus, Type 1

**DOI:** 10.3390/vaccines11040829

**Published:** 2023-04-12

**Authors:** Grant Klug, Frances M. Cole, Mark D. Hicar, Connie Watt, Tami Peters, Seth H. Pincus

**Affiliations:** 1Department of Chemistry and Biochemistry, Montana State University, Bozeman, MT 59717, USA; klug.grant@gmail.com (G.K.); fmycole@gmail.com (F.M.C.); cmwatt29@gmail.com (C.W.); tami.peters@montana.edu (T.P.); 2Department of Pediatrics, Jacobs School of Medicine and Biomedical Sciences, The University at Buffalo, Buffalo, NY 14203, USA; markhica@buffalo.edu

**Keywords:** HIV envelope, gp41, immunoconjugate, immunotoxin, CD4, cytotoxicity, persistent reservoir

## Abstract

We are developing cytotoxic immunoconjugates (CICs) targeting the envelope protein (Env) of the Human Immunodeficiency Virus, type 1 (HIV) to purge the persistent reservoirs of viral infection. We have previously studied the ability of multiple monoclonal antibodies (mAbs) to deliver CICs to an HIV-infected cell. We have found that CICs targeted to the membrane-spanning gp41 domain of Env are most efficacious, in part because their killing is enhanced in the presence of soluble CD4. The ability of a mAb to deliver a CIC does not correlate with its ability to neutralize nor mediate Ab-dependent cellular cytotoxicity. In the current study, we seek to define the most effective anti-gp41 mAbs for delivering CICs to HIV-infected cells. To do this, we have evaluated a panel of human anti-gp41 mAbs for their ability to bind and kill two different Env-expressing cell lines: persistently infected H9/NL4-3 and constitutively transfected HEK293/92UG. We measured the binding and cytotoxicity of each mAb in the presence and absence of soluble CD4. We found that mAbs to the immunodominant helix-loop-helix region (ID-loop) of gp41 are most effective, whereas neutralizing mAbs to the fusion peptide, gp120/gp41 interface, and the membrane proximal external region (MPER) are relatively ineffective at delivering CICs. There was only a weak correlation between antigen exposure and killing activity. The results show that the ability to deliver an effective IC and neutralization are distinct functions of mAbs.

## 1. Introduction

The reservoir of cells carrying functional HIV proviruses that persist during effective antiviral therapy (ART) is the barrier to curing HIV infection [1,2]. Patients may be maintained with undetectable viral loads for years on ART, a state termed clinical latency. But when ART is stopped, viremia rapidly returns. This is due to the retention of a functional provirus, viral sequences integrated as DNA into the genome of long-lived T lymphocytes. HIV utilizes the same activation signals as the T-cell, so immune stimulation also results in virus production. If this occurs on ART, the infection is contained by the drugs. Thus, patients require continuous antiviral therapy. The degree of HIV transcription in cells of the persistent reservoir during clinical latency is a matter of considerable study [1,3,4,5,6]. The degree of virus transcription within this reservoir is a key factor in considering approaches to the eradication of the persistent reservoir. Some approaches do not require virus transcription because the therapeutic agent can inactivate the virus and not interfere with the function of uninfected cells. These include: 1. genetic manipulations, including CRISPR-mediated gene inactivation; 2. enhancing immune responses via the administration of engineered mAbs, therapeutic immunization, and checkpoint inhibition; and 3. the use of drugs that specifically inactivate HIV [7,8,9,10]. Other approaches require some degree of virus transcription to target the therapies to cells that express the virus. If this does not occur spontaneously, then it may be necessary to pharmacologically enhance the expression of virus, the fancifully named “kick and kill” or “activate and purge” approaches to eradicate the persistent reservoir of infection [11,12,13]. 

In a kick and kill regimen, patients would be treated with latency-reversing agents to activate expression of virus proteins while preventing the spread of infection by continued (or even intensified) ART. Multiple different agents have been proposed as latency-disrupting agents, including histone-deacetylase inhibitors, non-canonical activators of NFκB signaling, SMAC mimetics, interleukins, and toll-like receptor agonists [14,15,16,17,18,19,20,21,22,23]. Many of these agents have been tested in humans and macaques and have been found to be effective in increasing virus transcription. However, a major concern remains whether the entire reservoir can be activated to the level necessary to result in its elimination. Once the virus can be induced to come out of hiding, proposals for purging the exposed cells encompass a variety of methods. Initially, removal of activated cells depended upon host immune response and/or viral cytopathic effect. When this proved ineffective, attention shifted to the use of mAbs and therapeutic immunization in combination with virus activation [18,21,22,23,24]. Promising results have been obtained in macaques. Presumably, the mechanisms of action of mAb-based therapies are a combination of neutralization and Fc-mediated functions, such as complement activation and antibody dependent cellular cytotoxicity (ADCC). We and others have put our efforts into the development of cytotoxic immunoconjugates (CICs) as a more effective approach to killing cells that express HIV proteins [25,26,27,28,29,30,31,32,33,34,35,36,37,38,39,40,41], demonstrating the ability of an immunotoxin to kill target cells that are not killed by complement [27]. CICs contain two domains: a targeting domain that binds to the cells marked for killing and the toxic domain. If the toxic moiety is a small drug, the agent is termed an antibody–drug conjugate; if a protein toxin is used, it is an immunotoxin. By combining a highly specific targeting moiety with an extremely toxic molecule, CIC’s can achieve effective killing of target cells with surprisingly little off-target toxicity. The in vivo efficacy of immunotoxins directed to the gp41 domain of the HIV envelope protein (Env) has been demonstrated in mice and macaques [37,41].

In the studies presented here, we focus on optimizing the targeting domain, anti-gp41 mAbs. Env is the sole HIV protein expressed intact on the surface of infected cells. We have previously examined large panels of anti-Env mAbs to identify those most effective at delivering CICs, and we identified the immunodominant (ID)-loop of gp41 as the most effective target [34,40,42]. The efficacy of gp41-targeted CICs is further enhanced by the addition of soluble CD4 (sCD4), which increases both gp41 exposure on the cell surface and internalization of Env [32,34,40]. Here we examine a panel of 30 anti-gp41 mAbs for binding to and killing Env-expressing cells in the presence and absence of sCD4. We used flow cytometry to quantify binding and an indirect immunotoxin assay to measure cytotoxicity. We again demonstrated the primacy of the ID-loop region of gp41 as a target for CIC killing. We found little correlation between CIC and neutralization targets. A minimum degree of binding to cell-surface Env was required for cytotoxicity; beyond that, there was little correlation between cell-surface antigen expression and cell killing.

## 2. Materials and Methods

### 2.1. Reagents and Cells

The mAbs used in these studies and their sources are listed in Table 1; the mAbs are displayed based upon epitope specificity, with those targeting the amino terminus shown at the top of the table, and descending moves towards the C-terminus, although there are anomalies due to conformational epitopes. In addition to these mAbs, we have previously tested other anti-gp41 for delivering immunotoxins to HIV-infected cells, including 12.6D, 15.5A, 16.11A, 2.2B, 41.1, 41.4, T15G1, and 8ANC195 [34,40,42].

Soluble two-domain CD4_183_ (sCD4) was obtained from the Aids Reagent Program [43]. The secondary immunotoxin consisted of affinity purified goat anti-human IgG (H + L chain specific, Invitrogen, Waltham, MA, USA) conjugated via the disulfide-cleavable heterobifunctional cross-linking reagent SPDP (Pierce Chemical Company, Rockford IL, USA) to deglycosylated ricin A chain at a 2–3 fold molar excess of ricin to antibody, as described elsewhere [40,44]. The deglycosylated ricin A chain was the kind gift of Ellen Vitetta. The same polyclonal Ab was purchased from Invitrogen conjugated to FITC. Two cell lines expressing HIV Env were used in these studies. H9/NL4-3 cells were infected with the molecularly cloned HIV isolate NL4-3, which utilizes Env from the prototype lab-adapted HIV isolate IIIB [45] and were the gift of Bruce Chesebro and Kathy Wehrly at NIAID [31]. Through serial passage, >99% of H9/NL4-3 cells maintain production of infectious virus and expression of cell surface Env [40]. 293T/92UG cells are HEK-293T cells transfected to express the Env of the clade A clinical isolate 92UG037.8 and were the kind gift of Bing Chen, Boston Children’s Hospital and Harvard Medical School [46]. Cell-surface Env on 92UG cells is found in its native trimeric configuration. Cells were cultured in RPMI 1640 (H9/NL4-3) or DMEM (92UG) supplemented with 10% fetal bovine serum (Hyclone) and grown at 37° in a 5% CO_2_ humidified atmosphere.

### 2.2. Flow Cytometry to Detect Cell-Surface Env Expression

Cells were cultured in the presence or absence of sCD4 (500 ng/mL) for 24 h before harvest for flow cytometry. Cells were washed twice in PBS with 1% bovine serum albumin and 0.01% Na azide (PBA), and 10^5^ were placed in 100µL in 96-well round bottom plates and incubated with 3µg/mL of anti-gp41 mAb with or without sCD4 (500 ng/mL) for 1 h at room temperature. Cells were washed twice with PBA and incubated with 1 µg/mL of FITC-conjugated goat anti-human IgG. Following a 1 h incubation at room temperature in the dark, the cells were washed twice and fixed in 2% paraformaldehyde. Using an Accuri C6 flow cytometer (BD Biosciences, San Jose, CA, USA), we analyzed 10^4^ events gated by forward and side scatter to exclude debris using FloJo software (Treestar/BD Biosciences). Data are presented as the median fluorescence intensity ± standard error of the mean.

### 2.3. Indirect Immunotoxin Assay

The ability of mAbs to deliver a cytotoxic payload was measured using an indirect cytotoxicity assay, as described previously [40,42,44]. The specificity of this assay has been demonstrated with target cells not expressing cell-surface Env (H9 and 293T) and with irrelevant mAbs. Target cells (1–2 
×
 10^4^, depending on cell type) were plated in triplicate in 96-well flat bottom tissue culture plates in 100 µL tissue culture medium in the presence of the indicated concentration of mAb, plus or minus sCD4 (500 ng/mL) for 1 h at 37°. As a negative control in every experiment, wells in the absence of mAb were included. The secondary immunotoxin, goat anti-human IgG conjugated to ricin A chain, was then added at a final concentration of 300 ng/mL in 100 µL, and the cells were incubated for 3 days. Wells receiving the secondary immunotoxin but not mAb serve as the negative control and are used in the calculation of percent cytotoxicity. MTS/PMS substrate (CellTiter AQueous, Promega) was added to the wells for the final 3 h of incubation. At the completion of incubation, absorbance (A) at 490 nm was read on a microplate reader (BioTek). Percent cytotoxicity was calculated according to the formula:{1−[(A_mAb_ − A_no cells_)/(A_no mab_ − A_no cells_)]} × 100. 

Results are shown as mean and SEM. If error bars are not visible in the figures, the error is smaller than the symbol.

### 2.4. Statistical Analyses

Flow cytometry was analyzed using FloJo’s built-in statistical packages. GraphPad Prism v8.3 (GraphPad Software, Boston, MA, USA) was used for Pearson correlation analysis and construction of figures. 

## 3. Results

### 3.1. Binding and Cytotoxicity of Anti-gp41 on Cells Transfected to Express Cell-Surface Env

HEK293T cells transfected to express Env of the clade A isolate 92UG037.8 (henceforth 92UG) were the kind gift of Dr. Bing Chen. The mAbs were tested for binding to 92UG by flow cytometry and for cytotoxicity with an indirect immunotoxin assay (using anti-human IgG conjugated with ricin A chain). We have used this assay previously and shown it to be predictive of which mAbs will function best when directly conjugated to a toxin or cytotoxic drug [40,44]. Testing was performed in the presence or absence of sCD4, which has been shown to increase exposure and internalization of gp41 epitopes, as well as enhance the cytotoxicity of gp41-targeted CICs [32,34,40,42]. Cell-surface expression of all epitopes was markedly increased in the presence of sCD4, particularly in the ID-loop region (Figure 1A). The exception to this was mAb 35O22, which recognizes the gp120/gp41 interface. The diminished expression of this epitope reflects the ability of sCD4 to dissociate gp120 and gp41 [70]. Cytotoxicity in the indirect immunotoxin assay is shown in Figure 1B. Previously published experiments with isotype controls and untransfected HEK293 cells have demonstrated that cytotoxicity in this assay is Env-specific [40,42]. With only a single exception, M44, all mAbs were able to deliver a cytotoxic signal. At this relatively high concentration of mAb (125 ng/mL), some mAbs required sCD4 to produce or enhance cytotoxicity, while others were equally effective with or without sCD4. Surprisingly cytotoxicity of mAb 4E10 was depressed by sCD4. In Figure 1C, we show that there was no correlation between epitope expression and cytotoxicity. Because we tested the mAbs for cytotoxicity at a relatively high concentration, we chose the most effective mAbs and titrated them in an indirect immunotoxin assay in the presence of sCD4 (Figure 2). The mAb concentration that resulted in 50% killing (IC_50_) of 92UG ranged from 0.2 to 1 ng for the seven most effective mAbs, all of which target epitopes in the ID-loop domain.

### 3.2. Binding and Cytotoxicity of Anti-gp41 mAbs on Persistently Infected Cells

H9/NL4-3 cells are a CD4+ lymphoma cell line persistently infected with HIV, with virtually all cells expressing Env on the cell surface and secreting infectious HIV [40]. Figure 3 shows the binding and cytotoxicity of the anti-gp41 mAbs when tested on H9/NL4-3 cells. Flow cytometry demonstrated that almost all mAbs bound the cells and that sCD4 enhanced the expression of all epitopes, especially those in the ID-loop region. The mAb to the gp120/gp41 interface, 35O22, again showed decreased binding in the presence of sCD4. The set of mAbs capable of delivering a cytotoxic payload to H9/NL4-3 cells was more restricted than we observed with 92UG. Killing was primarily observed with mAbs to the ID-loop region and only in the presence of sCD4. For 2C6, mutation of the framework three region (2C6-Fr3mut) diminished binding and abrogated cytotoxicity. In published findings, this mutant diminished binding and has less ADCC in comparison to 2C6 [52]. Contrary to what we observed in 92UG, cytotoxicity was well correlated with cell-surface epitope expression in the presence of sCD4. Figure 4 shows a dose–response curve on H9/NL4-3 and demonstrates differences between the cell lines used as targets. The IC_50_ of the most effective mAbs was roughly 10-fold higher on H9/NL4-3 (8–15 ng/mL) than on 92UG. We have consistently observed that 92UG are more sensitive to immunotoxin killing than H9/NL4-3 [40,42], and attribute this to a diminution of the effective dose received by the cells, resulting from the secretion of immunotoxin-coated virus by the infected H9/NL4-3, but not by the transfected 92UG.

Cytotoxicity of anti-gp41 mAbs with concentrations from 2 to 125 ng/mL was evaluated with ricin A chain conjugated anti-human Ig in the presence of sCD4183 (500 ng/mL). 

## 4. Discussion and Conclusions

We have compared the ability of human mAbs targeted to different regions of the transmembrane domain of the HIV envelope protein, gp41, to kill two different target cells that stably express HIV Env on the cell surface. For studies such as those performed here, stable expression of Env is necessary to allow for repetitions and comparison. H9/NL4-3 is unique among persistently HIV-infected cell lines in that it maintains a productive infection with no cytopathic effect and no need to periodically replenish the cultures with uninfected cells [40]. Immunotoxin-resistant variants that have lost expression of the cell-surface envelope are found at a frequency <10^−3^ [30,71,72]. The ability to maintain a high level of persistent infection results from a virus mutation (8AA deletion in Vpr), perhaps combined with a cellular mutation [71,73]. It has also been surprisingly difficult to transfect cells to stably express native trimeric envelope at reasonable levels, and the HEK293T/92UG cells are a rare example of such a cell line [46]. These cell lines do not necessarily express gp120/gp41 to the same degree as the relevant cells in patients. We and others have shown that directly conjugated immunotoxins effectively treat tissue culture infections of primary human T cells [34,41,74], although even these cells may or may not be representative of what occurs in patients. In this regard, the evidence of in vivo efficacy in experimental animals [37,39,41,75] is our best argument that CICs kill cells with relevant amounts of cell-surface Env expression. The utility of H9/NL4-3 and 92UG cell lines for these experiments lies in their stable expression of native Env without cytopathic effects, not that they reflect what occurs clinically.

The ability of a mAb to deliver a cell-killing moiety to a target cell differs from other Ab-mediated effector functions in that internalization of the CIC is necessary for it to function. ADCC, complement activation, and neutralization occur at the cell or virion surface. Previous studies, as well as these (Figure 1 and Figure 2 compared to 3 and 4), demonstrate that H9/NL4-3 are more resistant to CIC-directed cytotoxicity than 92UG [41]. We attribute this to the secretion of virus by H9/NL4-3 acting as an immunologic decoy, binding the CIC before it can reach the cell. We also observe here that once above a certain threshold binding, there is little correlation between binding of mAb to the cell, measured by fluorescence, and killing mediated by the same mAb. Epitope location is an important element in determining CIC-efficacy, and thus a weaker-binding mAb to a more effective epitope could yield greater killing. We have previously found that mAbs to the ID-loop make the most effective CICs and further confirmed this in the present studies. While multiple CICs were capable of inducing cytotoxicity in 92UG cells, the more resistant infected H9/NL4-3 cells were most effectively killed by ID-loop specific CICs, and only in the presence of sCD4. Because the ID-loop is highly mobile, it was the last domain to be visualized by cryo-EM or crystallography of Env and lies close to the cell surface [76,77]. We believe that both the mobility and proximity to the cell surface contribute to the effectiveness of CICs in this region, with CD4-mediated conformational shifts in Env initiating the first step in the internalization of the CIC [70,78,79]. 

There are two well-defined regions of gp41 that are targets of Ab-mediated neutralization: the N-terminal fusion peptide and MPER (Table 1). When tested for cytotoxicity on H9/NL4-3 cells, mAbs to these regions have consistently tested negative [34,40,42]. The neutralizing activity of ID-loop-specific mAbs is questionable, although recent studies have shown that mAbs to this region can prevent virus from infecting cells expressing FcγR1 (CD64) [80]. Nevertheless, among anti-gp41 mAbs, the epitope specificity of mAbs that neutralize is distinct from those that deliver CICs. Antibodies that mediate ADCC and those that can effectively deliver CICs to infected cells form partially overlapping subsets, with mAbs to the ID-loop providing both functions, whereas some anti-MPER mAbs mediate ADCC well but are ineffective in delivering CICs (Table 1, Figure 3). These observations regarding differences among anti-gp41 mAbs in neutralization, ADCC, and CIC function are consistent with earlier studies of epitopes on both gp41 and gp120 [34,40,42]. The fusion peptide is a relatively newly proposed target on gp41, and a few of the antibodies tested herein target that region. A more thorough evaluation of that region as new antibodies emerge may be warranted. 

We have demonstrated the in vivo efficacy of CICs targeted with mAb 7B2 in both mice and macaques [39,41] and utilized this mAb to construct antibody–drug conjugates as well as immunotoxins [41]. This mAb identifies a highly conserved gp41 epitope, the external loop CSGKLIC. In comparative titrations 7B2 and F240 [59,60] were most effective in killing both 92UG and H9/NL4-3 cell lines (Figure 2 and Figure 4). Future development and testing of anti-HIV CICs will continue to utilize mAb 7B2 to deliver cytotoxic payloads. We are currently testing 7B2-ricin A chain in HIV-infected humanized SCID mice. We acknowledge that it is unlikely this particular immunotoxin would be used in humans due to its immunogenicity. We have worked to reduce the immunogenicity of ricin A chain by pegylation, with only minimal success [41]. We are now focusing our efforts on identifying small drugs to be used in antibody drug conjugates. Unlike in cancer, where the target is, by definition, a dividing cell, the HIV reservoir likely contains non-dividing cells, even after viral activation [1,5,6]. Toxins kill resting cells, but small cytotoxic drugs developed primarily for cancer therapy may or may not. We have now screened a growing panel of small drugs used in anti-cancer conjugates for their ability to kill resting lymphocytes. We have conjugated the identified drugs to 7B2 and are now testing them in vitro for specificity and efficacy.

## Figures and Tables

**Figure 1 vaccines-11-00829-f001:**
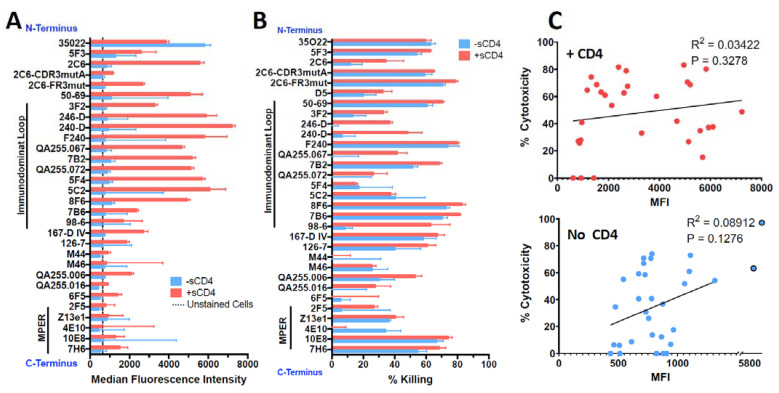
Comparison of binding and cytotoxicity of anti-gp41 mAbs assayed on 92UG cells. MAbs are shown ordered based on epitope location from N-terminus (**top**) to C-terminus (**bottom**). MAbs to MPER and immunodominant loop region are marked by bars. Data in red indicates presence of sCD4183 (500 ng/mL), blue the absence of sCD4183. (**A**) Indirect fluorescence. Mabs were tested at 3 µg/mL and detected with goat anti-human IgG-FITC as measured by flow cytometry. Binding is reported as median fluorescence with SEM. The vertical black line represents non-specific fluorescnce observed in the presence of anti-human FITC only. (**B**) Cytotoxicity. Anti-gp41 mAbs were tested at 125 ng/mL, and cytotoxicity was detected with ricin A chain conjugated anti-human Ig. Percent cytotoxicity was calculated as described in the text and shown as mean and SEM. (**C**) Correlation of cytotoxicity and binding with (**top**) and without (**bottom**) sCD4. P and R^2^ from Pearson correlation analysis.

**Figure 2 vaccines-11-00829-f002:**
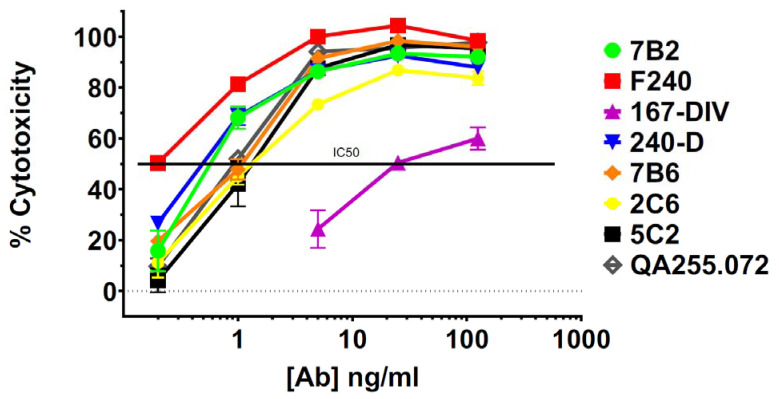
Titration of cytotoxicity of selected anti-gp41 mAbs assayed on 92UG cells. Cytotoxicity of anti-gp41 mAbs with concentrations from 0.2 to 125 ng/mL was evaluated with ricin A chain conjugated anti-human Ig in the presence of sCD4 (500 ng/mL). Percent cytotoxicity was calculated as described in the text.

**Figure 3 vaccines-11-00829-f003:**
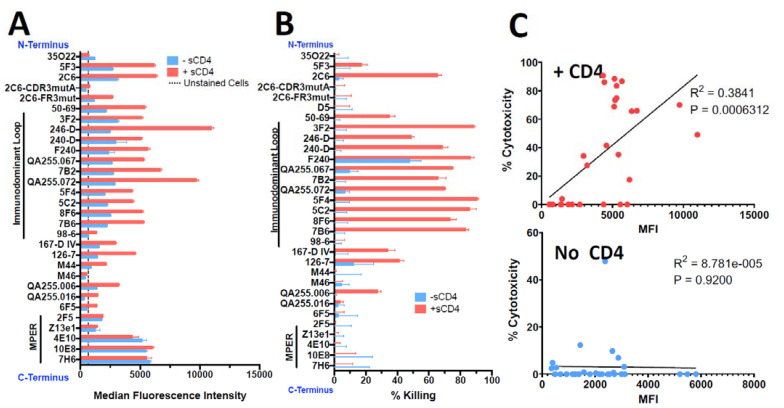
Comparison of binding and cytotoxicity of anti-gp41 mAbs assayed on H9/NL4-3 cells. H9/NL4-3 cells are H9 cells persistently infected with the lab isolate NL4-3. The layout of figures and reagents are as in the caption of Figure 1. (**A**) Indirect fluorescence, (**B**) Cytotoxicity, (**C**) Correlation of cytotoxicity and binding.

**Figure 4 vaccines-11-00829-f004:**
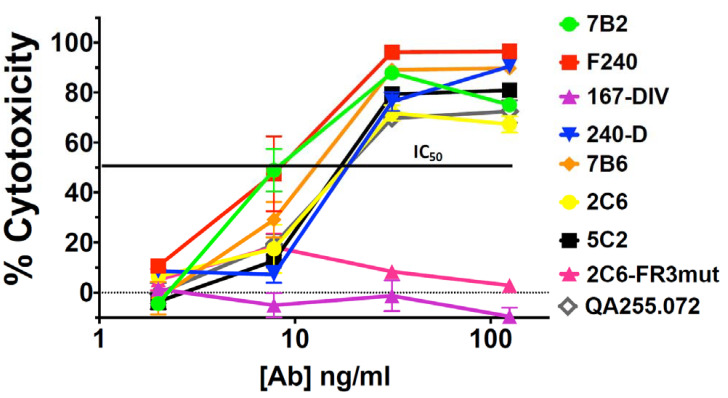
Titration of cytotoxicity with anti-gp41 mAbs assayed on H9/NL4-3 cells.

**Table 1 vaccines-11-00829-t001:** Monoclonal Antibodies Used in These Studies.

Antibody	Epitope Specificity (HXB2 Numbering)	Target Region ^a^	ADCC ^b^	Neutralize ^c^	Source	Citation
35O22	conformation (glycan + AA 88, 230, 241, 625)	gp120/41; Glycans	Weak	BN	ARP ^d^	[47,48]
5F3	conformational (524–534), (650–657)	FPPR/CHR		None	ARP	[49,50,51]
2C6	conformational (524, 592, 596)	ID loop; FP	Yes	none	M. Hicar	[52,53,54]
2C6-CDR3mutA	conformational (524, 592, 596)	ID loop; FP			M. Hicar	[52]
2C6-FR3mut	conformational (524, 592, 596)	ID loop; FP	Yes < 2C6		M. Hicar	[52]
50–69	conformational (579–613)	ID loop	Yes	none	ARP	[28,52,55,56,57]
3F2	585–597	ID loop			M. Hicar	[54]
246-D	590–597	ID loop	Yes	none	AR	[28,56,57,58]
240-D	592–600	ID loop	Yes	none	ARP	[56,58]
F240	592–606	ID loop	Yes	none	ARP	[59,60]
QA255.067	592–606	ID loop	Yes	none	J Overbaugh	[49]
7B2	596–606	ID loop	Weak	none	B. Haynes, J Robinson	[37]
QA255.072	596–609	ID loop	Yes	none	J Overbaugh	[49]
5F4	conformational (596, 598, 600, 604)	ID loop	Moderate	none	M. Hicar	[53,61]
5C2	conformational (596, 598, 600, 602, 604)	ID loop	Moderate	none	M. Hicar	[53,61]
8F6	conformational (596, 598, 600, 602, 603, 604)	ID loop	Moderate	none	M. Hicar	[53,61]
7B6	conformational (596, 598, 600, 604)	ID loop	Moderate	none	M. Hicar	[53,61]
98-6	conformational (579–613), (644–663)	ID loop; Cluster II	Weak	none	ARP	[28,52,56,57,58,59]
167-D IV	644–663	Cluster II	None	none	ARP	[56,62]
126-7	conformational (641–648)	Cluster II	Weak	none	ARP	[56,57,58,59]
M44	conformational	CHR/ID loop		BN	D. Dimitrov	[63]
M46	conformational	CHR/NHR		BN	M Dimitrov	[63]
QA255.006	conformational	CHR/FPPR/NHR	Yes	none	J Overbaugh	[49]
QA255.016	conformational	CHR/FPPR/NHR	Yes	none	J Overbaugh	[49]
6F5	conformational (557, 654, 657)	CHR/NHR	Yes	none	M. Hicar	[53,61,64]
2F5	662–668	MPER	Yes	BN	ARP	[65,66]
Z13e1	666–677	MPER		BN (	ARP	[67,68]
4E10	671–676	MPER	None	BN	ARP	[47,69]
10E8	671–683	MPER	Yes	BN	ARP	[47,59]
7H6	671–683	MPER	Moderate	BN	ARP	[47,48]

^a^ FPPR, Fusion peptide proximal region; CHR, C-terminal heptad repeat; ID loop, Immunodominant helix-loop-helix; FP, fusion peptide; NHR, N-terminal heptad repeat; MPER, Membrane proximal external region. ^b^ ADCC, Antibody dependent cellular cytotoxicity. ^c^ BN, broadly neutralizing. ^d^ ARP: AIDS Reagent Program, Germantown MD.

## Data Availability

All raw data upon which the figures in this manuscript are based are available upon request from Pincus.

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
