# Peer review of "Identification of Anti-gp41 Monoclonal Antibodies That Effectively Target Cytotoxic Immunoconjugates to Cells Infected with Human Immunodeficiency Virus, Type 1"

_vaccines, 2023, doi:10.3390/vaccines11040829_

Round 1
Reviewer 1 Report
In the study by Klug et al., the authors identify the potential anti-gp41 (HIV) antibodies for targeting HIV-infected cells. They have tested 30 antibodies that target the ID loop of gp41 with and without soluble CD4, which is known to expose gp41. They have tested the hypothesis that the cytotoxic anti-HIV immunoconjugates can be targeted to persistently infected cells. They compared the delivery efficiency of these Abs and found that toxicity by immunoconjugates is functionally different from the neutralization potential. While these observations in part confirm previous studies, the targeted killing of HIV-persistent cells remains to be achieved.
On the whole, the data appear to be diligently obtained, are transparently described and are an important contribution to the complex area of HIV-1 therapy. It would be helpful to test these antibodies on a T cell line expressing a physiological level of HIV env and CD4 receptor. If the experiment can’t be done, please discuss your results in detail in this respect. The discussion could be expanded into more general considerations about the difficulties in defining achieving targeted delivery to HIV-infected cells and other technologies for the same with respect to achieving a cure for HIV-1 infection.
Batra, H., Zhu, J., Jain, S., Ananthaswamy, N., Mahalingam, M., Tao, P., et al. (2021). Engineered Bacteriophage T4 Nanoparticle as a Potential Targeted Activator of HIV-1 Latency in CD4+ Human T-Cells. bioRxiv 2021, 453091. doi: 10.1101/2021.07.20.453091
Reviewer 2 Report
The manuscript by Klug, Cole and colleagues describes the use of antibodies against gp41 to target a cytotoxic antibody (goat anti-human IgG conjugated to ricin), to determine which antibodies are most effective at targeting HIV-1 env expressing cells. The goal is to develop a mechanism/therapy to purge persistent reservoirs of viral infection once HIV expression has been triggered by one of several proposed ways.
The results described indicate that antibodies against the helix-loop-helix region of gp41 were most effective, and that neutralizing antibodies were less effective. Unsurprisingly, the epitopes in the immunodominant loop were most enhanced by the pre-use of sCD4. Interestingly, there was no clear correlation between binding and cytotoxicity in 92UG cells, but it correlated well when the H9 cells were used.
The manuscript is relatively straightforward. I am not sure that this approach will necessarily be clinically useful, even when the cytotoxin is directly coupled to the anti-HIV antibody, but it is appropriate to explore it in vitro.
1. Perplexingly, negative controls (cell line not expressing env, non-gp41 antibodies, etc.) were not included in the experiments, and instead are addressed as having been produced in a separate manuscript. I am not sure this is adequate, as the background level of cytotoxicity of the goat anti-IgG should be subtracted from the results in order to be correct.
2. The choice of env expressing cells should also be discussed. Were they samples of convenience? Were other cells also tried? The H9 cells express a laboratory isolate, whereas the 92UG cells express a clinical isolate.
3. The marked difference in discordance/concordance between binding and potential cytotoxicity depending on which cell line/env is used is also unusual. It is not addressed in the discussion or results section; it leads to questions about the reliability of this assay.
Round 2
Reviewer 2 Report
The manuscript has been revised and my principal concerns - particularly the inclusion of negative controls - have been addressed by modification of the manuscript and the figures.